# AMR Parsing is Far from Solved:
# GrAPES, the Granular AMR Parsing Evaluation Suite

**Jonas Groschwitz**
University
of Amsterdam
j.groschwitz@uva.nl

**Shay B. Cohen**
University
of Edinburgh
scohen@inf.ed.ac.uk

**Lucia Donatelli**
Vrij Universiteit
Amsterdam
l.e.donatelli@vu.nl

**Meaghan Fowlie**
Utrecht
University
m.fowlie@uu.nl

## Abstract

We present the Granular AMR Parsing Evaluation Suite (GrAPES), a challenge set for Abstract Meaning Representation (AMR) parsing with accompanying evaluation metrics. AMR parsers now obtain high scores on the standard AMR evaluation metric Smatch, close to or even above reported inter-annotator agreement. But that does not mean that AMR parsing is solved; in fact, human evaluation in previous work indicates that current parsers still quite frequently make errors on node labels or graph structure that substantially distort sentence meaning. Here, we provide an evaluation suite that tests AMR parsers on a range of phenomena of practical, technical, and linguistic interest. Our 36 categories range from seen and unseen labels, to structural generalization, to coreference. GrAPES reveals in depth the abilities and shortcomings of current AMR parsers.

## 1 Introduction

Abstract Meaning Representation (AMR; Banarescu et al. 2013) parsing, the task of predicting a graph like the one in Fig. 1 for a given sentence, has improved by leaps and bounds in recent years. In fact, parsing performance of recent parsers, as evaluated by Smatch score (Cai and Knight, 2013), has reached and even surpassed reported human inter-annotator agreement (Bai et al., 2022; Banarescu et al., 2013). Has AMR parsing reached human performance, and thus has this form of semantic parsing been solved? Opitz and Frank (2022) perform human-expert evaluation for AMR parsing and find that, no, AMR parsing is far from solved: only 62% of parser outputs were rated acceptable.

In this work, we present an evaluation suite for English AMR parsing, including new data end metrics, that measures performance of AMR parsers with unprecedented breadth, detail and accuracy. Instead of computing a single score, like Smatch, on a single test set, we evaluate parsing performance on a range of phenomena of practical, tech-

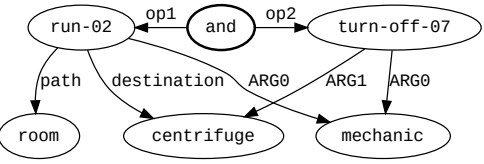

Figure 1: AMR for *The mechanic ran across the room to the centrifuge and turned it off.*

nical and linguistic interest. We answer the questions of how well parsers can handle pragmatic coreference, ellipsis, PP attachment, rare words, unseen named entities and structural generalization, just to name a few. Our Granular AMR Parsing Evaluation Suite (GrAPES) combines a selection of existing and novel sentence-AMR pairs.

A central theme in our development of GrAPES was that our metrics should actually evaluate what they promise to evaluate. To this end, we developed novel evaluation metrics specifically designed for the evaluation categories in our dataset. We also annotated, where necessary by hand, which graph-sentence pairs are correct and unambiguous, as well as which pairs are relevant for each category. We further annotated what part of each graph is relevant, reducing possible distractors.

Our work has three goals. First, to give quantitative results on a set of different phenomena in AMR parsing, so that we as a community know what abilities AMR parsers have, and where we can trust them to get it right. Second, to provide a tool that compares AMR parsers in more detail, and makes their differences visible. And third, to allow developers of AMR parsers to see where their systems still struggle – what needs to be improved. In this sense, GrAPES also functions as a challenge to the community. The experimental results we present in this paper confirm that GrAPES can serve these three purposes.

Our main contributions are:

- A practically and linguistically informed evaluation suite with 36 categories grouped into 9

sets

- Fine-grained evaluation metrics by category
- Evaluation results and statistical analysis for three recent parsers
- Detailed analysis tools for parser developers.

GrAPES is available open source at `https://github.com/jgroschwitz/GrAPES`.

We start by discussing related work in Section 2. We describe our categories and why we selected them in Section 3, and our measures to achieve high quality data and metrics in Section 4. We follow this up with an analysis of results for three recent AMR parsers (Section 5) and recommendations for how to use our evaluation suite (Section 6).

## 2 Related Work

Evaluation for AMR parsing is currently an active field of research. The standard evaluation metric to compare monolingual AMRs (both parsed and human annotated) is Smatch (Cai and Knight, 2013), which uses a graph matching algorithm to compute precision, recall, and F-score for semantic overlap of nodes and edges. Though designed to measure overall semantic adequacy of a predicted graph, recent advances in AMR parsing performance appear to have outgrown Smatch, spawning research on how to develop more fine-grained and interpretable evaluation metrics. Cai and Lam (2019), in addition to computing traditional Smatch on parser results, present variants of Smatch that emphasize 'core semantics' based on triple relations to the root, or predicative core, of the AMR graph. Similarly, Opitz and Frank (2022) show that Smatch often misses small but significant semantic divergences and recommend supplementing the metric with other metrics (including human analysis) to measure more fine-grained semantic adequacy. Opitz (2023) extends this work and separates Smatch into scores for pre-processing, alignment, and scoring. The above works reveal inconsistencies in how Smatch measures parser performance and point to the need for more focused evaluation methods.

Targeted evaluations for AMR parsing in the form of phenomenon-specific benchmarks aim to address this need. Szubert et al. (2020) investigate the role of reentrant edges specifically. Damonte et al. (2017) is the closest analogue of GrAPES, presenting an evaluation metric that pinpoints distinct subtasks of AMR parsing. They introduce a set of nine metrics that measure challenging linguistic phenomena such as reentrancies and named entities. We take this idea further, evaluating on 36 categories, expanding both breadth and depth. Moreover, we put additional emphasis on disentangling parsing performance on specific phenomena from overall performance, resulting in more precise and interpretable metrics. Finally, GrAPES includes newly annotated AMR-sentence pairs (both hand-built and grammar-generated) specifically designed to evaluate certain phenomena.

Beyond AMR evaluation, a growing literature on structural generalization is relevant to the AMR parsing task in developing evaluation suites for difficult semantic phenomena rooted in linguistic structure. For example, in the COGS dataset (Kim and Linzen, 2020) parsers must predict logical forms for sentences comprised of novel structural combinations, mimicking human ability to generalize compositionally. This dataset has recently been extended in the SLOG dataset for semantic parsing (Li et al., 2023), which targets more difficult structural phenomena using the same logical form.

GrAPES takes both AMR and non-AMR benchmarks as inspiration for a more comprehensive evaluation for AMR. We include several existing tasks into our dataset: (i) the Winograd Schema Challenge (Levesque et al., 2012), which consists of pairs of sentences with pronominal coreference that require pragmatics to disambiguate; (ii) *Putting words into BERT's mouth* (Karidi et al., 2021) for word disambiguation on simple sentences; and (iii) the Unbounded Dependencies Corpus (Rimell et al., 2009) which contains real life examples of long range dependencies in different categories.

## 3 Challenge Categories

We cover a broad range of phenomena that are interesting from practical, technical, and linguistic perspectives.

### 3.1 Four Example Categories

First, let us look at four of the categories in our dataset in some detail; they will illustrate our decisions in selecting categories below.

**Frequent predicate senses**: AMR builds on OntoNotes (Hovy et al., 2006) to disambiguate senses of predicates. For example in the AMR in Fig. 1, the "-02" suffix in run-02 specifies the sense to be "walk quickly", as opposed to e.g. "operate". We say such a predicate sense is *frequent* if it occurs at least 30 times in the AMR 3.0 training

set. To ensure that sense disambiquation is actually necessary, we also require that other senses for the same lemma in total occur at least 30 times. One of our categories tests parsing accuracy for such frequent predicate senses.

**Rare node labels:** The node label *centrifuge* in the example in Fig. 1 is *rare* in the training set (occurring up to 5 times). One of our categories measures a parser's ability to predict such rare labels.

**Pragmatic coreference:** AMR directly represents coreference in the graph: The fact that *it* refers to *centrifuge* in the sentence of Fig. 1 is represented with a *reentrancy*: both the destination-edge of run-02 and the ARG1 edge of turn-off-07 point to the same node. We include both pronominal and non-pronominal coreference. Coreference cannot be resolved by syntactic clues alone, and needs semantic and pragmatic information to resolve.[1] We measure parsing performance on this type of reentrancy in its own category, and include more categories for other types of reentrancy.

**Structural generalization for CP recursion:** CP (Complementizer Phrase) recursion, as in *You knew [that I said [that the men left]$_{CP}$]$_{CP}$*, can in principle have unlimited depth. Evaluating parser performance on sentences with particularly high CP recursion depth – higher than occurred in the training data, up to depth 10 – is one of our structural generalization categories.

## 3.2 Selecting Categories by Principle

To ensure that our 36 categories cover a diverse range of phenomena, we looked at our category selection through a selection of different lenses.

The first is the lens of sparsity. For some decisions a parser must make, such as the frequent predicate senses, the parser has plenty of training data. For other phenomena, such as rare node labels, the Zipfian distribution of language means that while the node labels themselves are each rare, *in total*, rare words are common. Finally, some phenomena are truly rare: the deep CP recursions in our structural generalization tasks feature nesting of the same grammatical structure to a depth that does not occur in the AMRBank at all. These truly rare phenomena are thus more of theoretical interest (but especially so).

The second lens is that we include both lexical

challenges (rare words, sense disambiguation, etc.) and structural challenges (pragmatic reentrancies, CP recursion, etc.).

Finally, we include a broad range of expected difficulty in our challenge categories. Some tasks are essentially impossible for current parsers, such as the predicate sense disambiguation for unseen senses (we explain why in Section 5). Some we expected to be difficult, such as deep CP recursions and pragmatic reentrancies. We also intentionally include some categories that we expect current parsers to perform well on, such as the sense disambiguation for frequent predicate senses, to check whether that expectation matches reality.

Table 1 shows all categories in GrAPES.

## 4 Dataset and Evaluation Design

With a wide range of phenomena selected, our guiding principles in creating the actual dataset and the evaluation metrics that form GrAPES are:

1. High annotation quality.
2. Metrics should measure the phenomenon they are supposed to measure, and nothing else.

### 4.1 Corpus Creation

Our four sources of data are the AMR 3.0 test set, other existing corpora, grammar-generated sentences, and hand-crafted sentences. Here we explain how we added them to GrAPES.

#### 4.1.1 AMRBank 3.0 Test Set

Most of the phenomena we test already occur at least to some extent in the test set of the AMR-Bank 3.0 (Knight et al., 2021). We extract relevant sentences for a range of our categories.

For each such category, a script extracts candidate corpus entries (sentence-AMR pairs) from the test set. E.g., for rare words, for every node label that occurs 1-5 times in the training set, we pull every entry in the test set with that node label.

We then manually filter the extracted dataset if necessary. This can have multiple reasons. First, some of the extracted examples have annotation errors. This is more frequent in some categories – for example, an unseen node label may be unseen simply because it is erroneous, and the corresponding word is not actually unseen, but has been annotated differently (correctly!) in the training set. We exclude such errors whenever feasible.

Other sentences are ambiguous, or the AMR guidelines do not fully specify what the correct

---

[1]An exception are second and first person pronouns – multiple mentions of *I* in a sentence refer to the same entity unambiguously. We measure these reentrancies in the separate category Unambiguous coreference.

**1. Pragmatic reentrancies**

Pragmatic coreference                      (T, C)
*__Obama__'s VP said __the president__ forgot __his__ coat.*

**2. Unambiguous reentrancies**

Syntactic (gap) reentrancies              (T)
*She __wants and needs__ to enter the room __whistling__*

Unambiguous coreference                   (T)
*__I__ raised __my__ fists in __self__-defence*

**3. Structural generalization**

Nested control and coordination           (G)
*The boy wanted to force the doctor to refuse to attend and jumped.*

Multiple adjectives                        (G)
*A strange big antique square dark container*

Centre embedding                           (G)
*The astronaut who [[the girl who the boy hugged] taught] left*

Long lists                                 (G)
*Please buy a book, gasoline, fish, expensive food, beer, soap, a map, a phone and coal.*

CP recursion                               (G)
*The lawyer said [that you knew [that the men mentioned [that the women left]]]*

CP recursion + coreference                 (G)
*I thought that __the doctor__ heard that the lawyer mentioned that the girls hated __her, the doctor__*

CP recursion + relative clause (RC)        (G)
*The __girls__ [who we claimed [that you thought [slept]$_{CP}$]$_{CP}$]$_{RC}$ __hated__ the lawyer*

CP recursion + RC + coreference            (G)
*The astronaut [who we said [liked the __lawyer__]$_{CP}$]$_{RC}$ actually hated __her__ after all*

**4. Rare and unseen words**

Rare node labels                           (T)
`centrifuge`

Unseen node labels                         (T)
`gown`

Rare predicate senses (excl. -01)          (T)
*Loose tee shirts* ⇒ `loose-03`

Unseen predicate senses (excl.-01)         (H)
*The young reporter filled in for the usual news anchors.* ⇒ `fill-in-07`

Rare edge labels (ARG2+)                    (T)
*We can get some commercial development* ⇒ `(develop-02 :ARG3 we)`

Unseen edge labels (ARG2+)                  (H)
*bounced onto the roof* `(bounce-01 :ARG4 roof)`

**5. Special entities**

Seen names                                 (T)
Unseen names                               (T)
`(name :op1 "Capitol" :op2 "Hill")`

Seen dates                                 (T)
Unseen dates                               (T)
`(date-entity :month 12 :day 22)`

Other seen entities                        (T)
Other unseen entities                      (T)
*...call him on his cell: 470-5715* ⇒ `phone-number-entity :value "470-5715"`

**6. Entity classification and linking**

Types of seen named entities               (T)
Types of unseen named entities             (T)
*LA* ⇒ `(city :name (name :op1 "LA"))`

Seen and/or easy wiki links                (T)
*North Korea* ⇒ `:wiki "North_Korea"`

Hard unseen wiki links                      (T)
*Zheng Chenggong* ⇒ `:wiki "Koxinga"`

**7. Lexical disambiguations**

Frequent predicate senses                  (T)
*He __used__ the tool* ⇒ `use-01`

Other word ambiguites                      (C, H)
*__in__ Canada* ⇒ `be-located-at-91`

**8. Edge attachments**

PP attachment                              (G)
*Sophie knew the journalist __with the telescope__*

Unbounded dependencies                     (C)
*I __love__ and hate paper __writing__*

Passives                                   (T)
*I was seen* ⇒ `(see-01 :ARG1 i)`

Unaccusatives                              (T)
*I fell* ⇒ `(fall-01 :ARG1 i)`

**9. Non-trivial word-to-node relations**

Ellipsis                                   (T)
*drive back and forth (two* drive *nodes)*

Multinode word meanings                    (T)
*baker* ⇒ `(person :ARG0-of bake-01)`

Imperatives                                (T)
*Go!* ⇒ `(go-02 :mode imperative :ARG0 you)`

Table 1: All categories in GrAPES, grouped into 9 sets. Letters in brackets are data sources: T = AMR testset, G = Grammar, H = Handcrafted for GrAPES, C = Other corpora (Levesque et al. (2012) for Pragmatic coreference, Karidi et al. (2021) for Ambiguous words, and Rimell et al. (2009) for Unbounded dependencies).

annotation for the sentence should be. We also exclude these sentences, since they can lead to false negatives (when the parser predicts one option, but the gold annotation is a different one).

In other cases, the heuristics by which we extract the candidates are not precise enough. For instance, our script that extracts reentrancies looks only for undirected cycles in the graph. We hand-annotate these sentences with their category of reentrancy: Pragmatic coreference (*Mary thinks Susan likes __her__*), Syntactic gap (*She wants to sleep*), or Unambiguous coreference (*I like __my__ hair*).

For any category, if initial sampling indicates that the rate of erroneous or ambiguous examples, or examples that do not fit the category, is above 10%, we filter the data by hand, selecting only correct, relevant examples of low ambiguity.

### 4.1.2 Other existing corpora

We include sentences from the Winograd Schema Challenge (Levesque et al., 2012), *Putting Words into BERT's Mouth* (Karidi et al., 2021) and the Unbounded Dependency Corpus for CCG parsing (Rimell et al., 2009) in our evaluation suite. However, none of these corpora had been annotated with AMRs. We add AMR annotations, or partial AMRs for the relevant subgraphs.

### 4.1.3 Generation from grammars

For structural generalization categories and for PP attachment, we write synchronous grammars that generate sentences and their graphs, using Alto (Gontrum et al., 2017), and sample from the language of the grammar. This gives us sentences at every desired recursion depth. For PP attachment, using a grammar allows us to add more lexical variety to sentences, while keeping them pragmatically unambiguous, e.g. *The professor observed the army with the binoculars; The baker looked at the moon with the spyglass*.

### 4.1.4 Hand-crafted

Finally, for some rare lexical phenomena, not enough relevant entries occur in the test set. For these we hand-crafted sentences and annotated them with graphs or partial graphs as necessary. We added short, simple sentences such as, for Unseen predicate senses, *The comedian has a dry sense of humor* (sense `dry-04`).

A detailed description on how we obtained the

corpus for each category is given in Appendix B.

## 4.2 Evaluation Metrics

To evaluate overall performance on a data set, Smatch is the standard for AMR. Smatch evaluates all phenomena by calculating the precision, recall and F1 for all triples in the graph (e.g. [source, edge label, target]). In our evaluation, however, we usually want to zero in on the phenomenon in question, ignoring other parts of the graph. For this we develop new evaluation tools, some of which we present in the following. A full list of metrics appears in Appendix B.

### 4.2.1 Metrics

Recall the AMR in Fig. 1 for the example (1).

(1)  The mechanic ran across the room to the centrifuge and turned it off.

**Node recall**: for many lexical phenomena, we check only whether a node exists in the predicted graph with the label in question; e.g. `centrifuge` in (1) for rare labels.[2]

**Edge recall**: Often we are interested in the presence of a particular edge. For instance, consider the prepositional phrase (PP) attachment of *to the centrifuge* – this PP here could, syntactically speaking, attach to the verb (*ran [. . . ] to the centrifuge*) or to the noun (*the room to the centrifuge*). To see if the attachment is correct, we check if there is any edge in the predicted graph between a node labeled `run-02` and a node labeled `centrifuge`, with the correct label and direction.

A parse that was correct except for drawing this edge wrong would have a Smatch score of 92, but on our measure would correctly get 0 for this metric on this entry. (That parse would, however, do fine for the Rare Words metric on this same entry.)

**Exact match**: In the Structural Generalization set of GrAPES, we designed the grammars such that there is little in the way of distractors, ambiguity, or lexical challenges. Moreover, by the nature of the task, the graphs are very schematic, with repeated structures. Failing to capture this repetition – that is, failure to capture this generalization – can be evident from a single misplaced edge, yielding a high Smatch score but poor generalization. For these sentences, therefore, we hold the parser to the high standard of exact match.

---

[2]In many metrics here we only use recall and not precision. Parsers "cheating" recall by predicting multiple labels in an attempt to hit the right one is not an issue we observed.

### 4.2.2 Prerequisites and Sanity Checks

Even the above phenomenon-specific metrics cannot always fully isolate the phenomenon, as we will see in the following. To further reduce false negatives, we use *prerequisites* and *sanity checks*.
**Prerequisites**: Consider a parse of (1) in which everything is right except the node label `centrifuge`, replaced by `machine`. Using edge recall to measure PP attachment as above, the edge in question is measured as being absent, since there is no edge between nodes labeled `run-02` and `centrifuge` – since there is no node labeled `centrifuge`. For our purposes, though, this should not count as failing at PP attachment, as the PP attachment is not the problem: the node label is. For this reason, for these kinds of metrics we also measure *prerequisites*: the parts of the graph that need to be present for the evaluation metric to be meaningful, but are not themselves what we look for. In our example, to measure the prerequisites, we check for the presence of nodes labelled `centrifuge` and `run-02`, because if these nodes do not exist in the first place, we cannot meaningfully evaluate the existence of an edge between them. For each metric, we can then use a parser's prerequisite score as that parser's ceiling for the phenomenon.
**Sanity Checks**: In Structural Generalization categories, we consider the whole graph, not just single edges and nodes. We therefore don't have prerequisites here. Instead, we use *sanity checks*. For most categories, these are unnested variants of the phenomena. For instance, for CP recursion, we check that a single CP can be embedded, as in *She thinks that they left*. In some, they are lexical checks, for instance in Long Lists, where we check for each item of a list separately. e.g. if *Please buy bread, eggs, and cheese* is in the generalization corpus, and the sanity check includes *Please buy bread*.

In total, GrAPES evaluates 19590 datapoints: 15441 from the AMR testset, 3643 from grammars, 307 from other existing corpora and 199 from handcrafted examples. In Tables 2 to 5, the rightmost column shows the number of datapoints for each metric for each category.

## 5 Performance of Current Parsers

### 5.1 Experimental Setup

To gain insights into the current state of the art in AMR parsing, we evaluate on GrAPES three recent parsers with very different parsing architectures. We evaluate:

| Set | Category | Metric | AM Parser | C&L | AMRBart | # |
|---|---|---|---|---|---|---|
| 1 | Pragmatic coreference (testset) | Edge recall | 06 [02, 18] | 08 [03, 22] | 39 [25, 55] | 36 |
| | | Prerequisites | 50 [34, 66] | 36 [22, 52] | 61 [45, 75] | 36 |
| | Pragmatic coreference (Winograd) | Edge recall | 02 [00, 13] | 05 [01, 17] | 32 [20, 48] | 40 |
| | | Prerequisites | 78 [62, 88] | 30 [18, 45] | 65 [50, 78] | 40 |
| 2 | Syntactic (gap) reentrancies | Edge recall | 24 [14, 39] | 24 [14, 39] | 49 [34, 64] | 41 |
| | | Prerequisites | 54 [39, 68] | 59 [43, 72] | 68 [53, 80] | 41 |
| | Unambiguous coreference | Edge recall | 10 [03, 25] | 39 [24, 56] | 65 [47, 79] | 31 |
| | | Prerequisites | 71 [53, 84] | 71 [53, 84] | 77 [60, 89] | 31 |

Table 2: Results on reentrancy categories. Gray numbers in square brackets are 95%-Wilson confidence intervals.

- The AM parser (Groschwitz et al., 2018), a neuro-symbolic compositional parser; we use the version with BERT embeddings, trained on AMRBank 3.0 (Lindemann et al., 2020).
- Cai and Lam (2020) (henceforth, C&L), a structured neural parser that iterates between analyzing the string and predicting the graph.
- AMRBart (Bai et al., 2022), a version of BART (Lewis et al., 2020) finetuned for AMR parsing with additional graph pre-training. We use the model trained on AMRBank 3.0.

Recall that we use the AMRBank 3.0 test set to extract some of our corpus, so in the following, "test set" always refers to that version. C&L was only trained on AMRBank 2.0, and we use that version here. Since AMRBank 2.0 is a subset[3] of AMRBank 3.0, all unseen/rare labels in 3.0 are still unseen/rare for 2.0; however, some of the parsing errors of C&L may be due to the training set, rather than the parsing architecture.

[3]There are also some annotation differences between the 2.0 and 3.0 versions, but not many: we compared the graphs in the testset of AMRBank 2.0 to their 3.0 counterparts and obtained a total Smatch score of 98 (out of 100), indicating nearly identical graphs.

## 5.2 Results

We report our metric results in Tables 2 to 5, and compactly in Table 7. We include 95%-Wilson confidence intervals (Wilson, 1927) in square brackets (gray) to give the reader an indication of the degree of uncertainty that results from the sample size.

We also include in GrAPES a tool for visualization of gold and predicted graphs. While our quantitative evaluation is already fine-grained, we believe that a qualitative evaluation of examples is crucial in interpreting the quantitative results.

In the following, we present some highlights of our evaluation, both qualitative and quantitative.

**The effect of sparsity.** Across the board, parsers struggle when little training data is available for the task, and the less training data they have available, the more they struggle. This applies for example to rare and unseen node labels (Table 4): the most recent parser, AMRBart, does not even get half of the unseen node labels right. Interestingly, the unseen node labels are the only category in which the older AM parser outperforms AMRBart, presumably because of the built in copy-mechanism, which AMRBart lacks. Similar patterns are vis-

| Set | Category | Metric | AM Parser | C&L | AMRBart | # |
|---|---|---|---|---|---|---|
| 3 | Nested control and coordination | Exact match | 48 [35, 61] | 08 [03, 19] | 36 [24, 50] | 50 |
| | Sanity check | Exact match | 100 [77,100] | 77 [50, 92] | 100 [77,100] | 13 |
| | Multiple adjectives | Exact match | 72 [57, 84] | 32 [20, 48] | 98 [87,100] | 40 |
| | Sanity check | Exact match | 100 [74,100] | 100 [74,100] | 100 [74,100] | 11 |
| | Centre embedding | Exact match | 30 [17, 48] | 13 [05, 30] | 57 [39, 73] | 30 |
| | Sanity check | Exact match | 85 [58, 96] | 100 [77,100] | 85 [58, 96] | 13 |
| | CP recursion | Exact match | 58 [48, 67] | 24 [17, 33] | 63 [53, 72] | 100 |
| | Sanity check | Exact match | 100 [61,100] | 100 [61,100] | 100 [61,100] | 6 |
| | CP recursion + coreference | Exact match | 01 [00, 04] | 09 [05, 14] | 46 [39, 53] | 182 |
| | Sanity check | Exact match | 29 [15, 49] | 29 [15, 49] | 88 [69, 96] | 24 |
| | CP recursion + relative clause (RC) | Exact match | 17 [09, 28] | 00 [00, 06] | 17 [09, 28] | 60 |
| | Sanity check | Exact match | 75 [30, 95] | 25 [05, 70] | 75 [30, 95] | 4 |
| | CP recursion + RC + coreference | Exact match | 00 [00, 05] | 00 [00, 05] | 13 [07, 23] | 70 |
| | Sanity check | Exact match | 00 [00, 43] | 00 [00, 43] | 80 [38, 96] | 5 |
| | Long lists | Conjunct recall | 02 [02, 03] | 35 [33, 37] | 93 [92, 94] | 1872 |
| | | Conjunct precision | 93 [82, 98] | 57 [54, 60] | 98 [97, 98] | 45 |
| | | Unseen :opi recall | 00 [00, 01] | 00 [00, 01] | 74 [70, 78] | 408 |
| | Sanity check | Exact match | 97 [92, 99] | 81 [73, 87] | 99 [95,100] | 111 |

Table 3: Results on structural generalization.

| Set | Category | Metric | AM Parser | C&L | AMRBart | # |
|---|---|---|---|---|---|---|
| 4 | Rare node labels | Label recall | 62 [59, 66] | 56 [53, 60] | 69 [66, 73] | 676 |
| | Unseen node labels | Label recall | 60 [51, 68] | 50 [41, 59] | 45 [37, 54] | 117 |
| | Rare predicate senses (excl. -01) | Label recall | 36 [24, 49] | 11 [05, 21] | 45 [32, 58] | 56 |
| | | Prerequisites | 89 [79, 95] | 73 [60, 83] | 91 [81, 96] | 56 |
| | Unseen predicate senses (excl -01) | Label recall | 05 [01, 17] | 00 [00, 09] | 00 [00, 09] | 40 |
| | | Prerequisites | 88 [74, 95] | 90 [77, 96] | 85 [71, 93] | 40 |
| | Rare edge labels (ARG2+) | Edge recall | 10 [04, 23] | 20 [10, 35] | 35 [22, 50] | 40 |
| | | Prerequisites | 57 [42, 71] | 55 [40, 69] | 72 [57, 84] | 40 |
| | Unseen edge labels (ARG2+) | Edge recall | 08 [03, 22] | 14 [06, 29] | 33 [20, 50] | 36 |
| | | Prerequisites | 44 [30, 60] | 61 [45, 75] | 53 [37, 68] | 36 |
| 5 | Seen names | Recall | 86 [85, 88] | 91 [90, 92] | 94 [93, 95] | 1788 |
| | Unseen names | Recall | 68 [65, 71] | 60 [56, 63] | 76 [73, 79] | 910 |
| | Seen dates | Recall | 79 [73, 84] | 72 [66, 77] | 94 [90, 96] | 233 |
| | Unseen dates | Recall | 47 [40, 54] | 57 [50, 64] | 86 [81, 90] | 204 |
| | Other seen entities | Recall | 80 [75, 85] | 84 [79, 88] | 97 [94, 99] | 237 |
| | Other unseen entities | Recall | 74 [65, 82] | 33 [25, 42] | 78 [69, 85] | 109 |
| 6 | Types of seen named entities | Recall | 83 [81, 85] | 79 [77, 81] | 92 [90, 93] | 1628 |
| | | Prerequisites | 85 [83, 86] | 91 [89, 92] | 94 [93, 95] | 1628 |
| | Types of unseen named entities | Recall | 35 [32, 39] | 29 [25, 32] | 51 [47, 55] | 659 |
| | | Prerequisites | 59 [55, 63] | 54 [50, 58] | 70 [66, 73] | 659 |
| | Seen and/or easy wiki links | Recall | 70 [68, 72] | 82 [80, 84] | 87 [85, 88] | 2064 |
| | Hard unseen wiki links | Recall | 00 [00, 01] | 18 [14, 23] | 09 [06, 13] | 277 |

Table 4: Results for sets 4-6: rare and unseen words, and special entities.

ible when comparing the prediction of frequent (Table 5) and rare (Table 4) predicate senses. For example, for C&L, the recall drops from 81 to 11. Named entities (Table 4) show the same picture: unseen entities are consistently more difficult to handle than seen ones. While this trend is not unexpected, we quantify it in new detail for AMR parsing, and provide a consistent method for measuring progress.

**Where the state of the art does well.** The most recent parser we test, AMRBart, achieves very high recall (92 and higher) for all categories of seen entities and their classification into types (Table 4). Passives and unaccusatives as well as frequent predicate senses receive lesser, but still strong scores (Table 5). In structural generalization, AMRBart nearly aces the Multiple adjectives test (Table 3).

**Successes and struggles on contextual disambiguation.** While a parser's ability to make contextual decisions is tested in many of our categories, it is particularly highlighted in pragmatic coreference (Table 2), as well as word and attachment disambiguations (Table 5). We find that across the board, AMRBart shows noticeable improvements over the older parsers, and achieves a respectable performance. However, there is still much room for improvement. For example, on the pragmatic coreferences extracted from the test set, among the edges where the prerequisites are satisfied, AMRBart still gets about one third wrong; performance on Wino-

grad is even worse[4] (Table 2). PP attachment has a similar error rate. Even for one of the best categories here, Frequent predicate senses, among the labels where the lemma was correct (i.e. the prerequisite satisfied), AMRBart gets about 10% of the senses wrong (Table 5). Since such sense ambiguities are so frequent (about one per sentence in the test set), even this small error rate quickly adds up.

**Structural generalization.** Some of our structural generalization categories (Table 3) compare quite directly to the COGS (Kim and Linzen, 2020) and newly extended SLOG (Li et al., 2023) datasets. For example, Weißenhorn et al. (2022b) show that finetuning BART on COGS gives 0% exact match for their CP recursion category; here AMRBart obtains 63%. This may be because the AMR training set is more diverse than COGS (which is restricted on purpose). While there is still a leap from the realistic language in the AMR training set to the generalization examples here, the parser may have more data to make a generalization *from*.

Still, structural generalization is hard. We get less than 50% exact match in most categories, and a qualitative analysis shows that the performance drops with depth. For example, all but two successful parses on CP recursion + RC come from samples where there is only one CP. Surprisingly,

---

[4]We note that fine-tuned large language models reach a performance of over 95% on the Winograd Schema Challenge (Chowdhery et al., 2022). The lower performance here may be due to less powerful models, or due to additional difficulties in solving the task in the AMR format.

| Set | Category | Metric | AM Parser | C&L | AMRBart | # |
|---|---|---|---|---|---|---|
| 7 | Frequent predicate senses (incl -01) | Label recall | 81 [79, 83] | 81 [79, 83] | 86 [84, 88] | 1654 |
| | | Prerequisites | 92 [90, 93] | 91 [90, 93] | 94 [93, 95] | 1654 |
| | Word ambiguities (handcrafted) | Recall | 77 [63, 86] | 79 [65, 88] | 91 [80, 97] | 47 |
| | Word ambiguities (Karidi et al., 2021) | Recall | 75 [65, 82] | 76 [66, 83] | 88 [80, 93] | 95 |
| 8 | PP attachment | Edge recall | 53 [48, 59] | 43 [38, 49] | 66 [61, 71] | 325 |
| | | Prerequisites | 94 [91, 96] | 86 [81, 89] | 95 [93, 97] | 325 |
| | Unbounded dependencies | Edge recall | 35 [24, 47] | 32 [22, 44] | 45 [34, 57] | 66 |
| | | Prerequisites | 65 [53, 76] | 59 [47, 70] | 65 [53, 76] | 66 |
| | Passives | Edge recall | 55 [45, 66] | 60 [49, 70] | 76 [66, 84] | 83 |
| | | Prerequisites | 75 [64, 83] | 73 [63, 82] | 80 [70, 87] | 83 |
| | Unaccusatives | Edge recall | 50 [36, 64] | 69 [55, 80] | 71 [57, 82] | 48 |
| | | Prerequisites | 71 [57, 82] | 75 [61, 85] | 79 [66, 88] | 48 |
| 9 | Ellipsis | Recall | 03 [01, 15] | 39 [25, 56] | 55 [38, 70] | 33 |
| | | Prerequisites | 91 [76, 97] | 94 [80, 98] | 94 [80, 98] | 33 |
| | Multinode word meanings | Recall | 58 [44, 71] | 60 [46, 72] | 84 [71, 92] | 50 |
| | Imperatives | Recall | 34 [25, 45] | 43 [33, 55] | 66 [55, 75] | 76 |
| | | Prerequisite | 82 [71, 89] | 80 [70, 88] | 89 [81, 95] | 76 |

Table 5: Results for sets 7-9: lexical ambiguities, edge attachments and non-trivial word-node relations.

the compositional AM parser, which has done very well on COGS (Weißenhorn et al., 2022a), does not excel here. A possible reason for this may be that we use a version of the AM parser called the *fixed-tree decoder*, which performs better on AMR overall (Lindemann et al., 2020). Weißenhorn et al. (2022a) use the *projective decoder*, noting that it yields better generalization results.

One thing to note is that the different generalization categories have similar sentence lengths, but different parser performance. This shows that we do not just measure sentence length effects here.

For the generalization categories that also include coreference, our qualitative evaluation showed a form of parser bias, where male pronouns where more often successfully resolved than female pronouns; details in Appendix A.

**"Impossible" tasks.** Some tasks are not possible to do in the classic paradigm of simply training a model on the training data, because external information is required. An example of this is the Unseen predicate senses category (Table 4), because the numbers chosen for senses in OntoNotes (like the 02 in run-02 in Fig. 1) are arbitrary with respect to the actual meaning. That is, if the sense was not observed in the training data, the only way to relate the sense marker to the meaning is to look it up in OntoNotes, and a parser that does not use that external resource cannot perform the task. Consequently, all parsers we tested score near 0 here. Similar observations apply to Hard unseen wiki links. C&L and AMRBart use external tools for wiki links (Daiber et al., 2013; Wu et al., 2020), and therefore obtain a non-zero accuracy here.

The AM parser is by principle unable to han-dle pragmatic coreference, long lists, or ellipsis (Groschwitz, 2019). This reflects in the low scores in our corresponding categories (Tables 2, 3, 5).

**Further difficulties.** There is serious room for improvement for all tested parsers in Syntactic (gap) reentrancies (which include reentrancies due to e.g. control verbs and coordination), Unambiguous coreference (both Table 2), and Ellipsis and Imperatives (Table 5).

### 5.3 Evaluating GrAPES

We have now seen how a range of parsers perform on our evaluation suite. But we also want to examine to what extent we have reached the design goals of GrAPES in terms of granularity and whether our metrics measure exactly what they are supposed to. In particular, we compare GrAPES to the closest previous work, Damonte et al. (2017). Table 6 shows the metrics of Damonte et al. for the three parsers that we evaluated on GrAPES.

**Fine-grained categories matter.** First, we can see that using more fine-grained categories actually matters. For example, Damonte et al. use a single metric for wiki links ("Wikification"). We split this category into seen and (hard) unseen wiki links and show that parser performance on the two is very different (Table 4). Similarly, Damonte et al. have a single category for reentrancies. We show that parsers perform noticeably better on unambiguous reentrancies, compared to reentrancies that require pragmatic understanding to resolve (Tables 2, 7).

These more fine-grained evaluation insights are all the more relevant because improving parser performance for each of these phenomena may require a different approach. As we noted above, predict-

| Metric | AM Parser | C&L | AMRBart |
|---|---|---|---|
| Unlabeled | 77 | 78 | 86 |
| No WSD | 75 | 76 | 84 |
| Named Entities | 81 | 74 | 88 |
| Wikification | 71 | 80 | 79 |
| Negations | 60 | 70 | 73 |
| Concepts | 86 | 84 | 90 |
| Reentrancies | 56 | 62 | 73 |
| SRL | 73 | 74 | 83 |

Table 6: Damonte et al. (2017) metrics (AMR 3.0 test)

| | AM Parser | C&L | AMR Bart |
|---|---|---|---|
| Smatch on AMRBank 3.0 | 75 | 75 | 84 |
| 1. Pragmatic reentrancies | 04 | 07 | 36 |
| 2. Unambiguous reentrancies | 17 | 32 | 57 |
| 3. Structural Generalization | 32 | 18 | 59 |
| 4. Rare and unseen words | 30 | 25 | 38 |
| 5. Special entities | 73 | 66 | 88 |
| 6. Entity classification and linking | 47 | 52 | 60 |
| 7. Lexical disambiguation | 77 | 79 | 89 |
| 8. Edge attachments | 48 | 51 | 65 |
| 9. Non-trivial word-to-node relations | 32 | 48 | 68 |

Table 7: Compact GrAPES results table. Scores are averages over non-prerequisite, non-sanity-check scores. Note that this averages scores that are on the same 0-100 scale, but not necessarily the same metric.

ing unseen wiki links requires knowledge outside of the standard AMR training data, in contrast to recalling wiki links seen during training. In addition, not all methods are equally suited for pragmatic and syntactic reentrancies, as the limitations of the AM parser on pragmatic reentrancies show.

**Successfully targeted metrics.** Measuring parser performance for a specific phenomenon, disentangled from overall parser performance, is a challenge. For example, the "No word sense disambiguation (WSD)" metric of Damonte et al. computes Smatch score, ignoring OntoNotes predicate senses. The difference to the original Smatch score should then show the impact of WSD errors. However, for example for AMRBart, both scores are 84, and for the AM Parser, both are 75 – the WSD errors disappear during rounding. By contrast, we show that for both frequent (Table 5) and in particular for rare and unseen (Table 4) predicate senses, the parsers make measurable mistakes.

We also computed the Reentrancy metric of Damonte et al. on the Pragmatic coreference (Winograd) portion of GrAPES, and found that the AM Parser obtains 55/100. This is in stark contrast to the recall of 2% that we measure. In part this is due to Damonte et al. measuring all types of reentrancies, while we focus on the pragmatic coreferences that the Winograd dataset was built for. But also, for Damonte et al.'s metric, which measures Smatch on specific subgraphs related to reentrancies, it is difficult to say what "55/100" exactly means. The recall on exactly the reentrant edges relevant to the Winograd schema challenge, which we measure, is more intuitively interpretable.

**Prerequisites and sanity checks.** Our use of prerequisites and sanity checks further helps in making our metrics targeted, allowing us to pinpoint the actual error types. Compare, for example, the performance of AMRBart on PP attachment and Unaccusatives (both Table 5). The numbers for edge recall are quite close, 66 and 71 respectively. However, on PP attachment, AMRBart satisfies the prerequisites nearly perfectly at 95%, in contrast to the lower prerequisite percentage on Unaccusatives (79%). This allows us to conclude that, correcting for this difference in prerequisites, AMRBart does much better on Unaccusatives than PP attachment.

Furthermore, the high parser performance levels on many sanity checks for structural generalization indicate that the difficulty in those categories does not just lie in some of the lexical items we used in our grammars, but indeed in the structural generalization (Table 3). A qualitative analysis showed that most existing errors on the sanity checks are structural rather than lexical, indicating that even without deep recursion, the structures we test here are not trivial for current parsers.

## 6 Recommendations

For researchers in AMR parsing who want to show their parser's results on GrAPES, we recommend including the more compact Table 7 in the main paper, as well as highlighting results from specific fine-grained categories as applicable. A complete table, combining Tables 2 to 5, should be included in the appendix. We encourage users of GrAPES to look at example parser output, to contextualize the metrics. Our evaluation suite includes code for visualization as well as for computing all results (and generating tables) for novel parser output.

## 7 Conclusion

We have shown that state-of-the-art AMR parsers still struggle with various phenomena, including data sparsity, contextual ambiguity resolution and structural generalization. We provide a detailed evaluation suite with custom metrics to measure progress in these areas.

## Limitations

Our dataset is designed specifically for AMR. While parsers on other semantic parsing tasks may make similar errors to the ones that we document here, drawing conclusions from our results to the overall state of semantic parsing should be done only carefully. However, we hope that this work serves as inspiration for creating similar evaluation suites for other tasks, in particular in syntactic and semantic parsing. While most of our implementation work is not directly usable for other formalisms (such as our manual AMR annotations or the code to filter the AMRBank for specific phenomena), some of it could be used with some adaptation. In particular, our grammars are built with Alto (Gontrum et al., 2017), which is specifically designed for multi-formalism grammars, meaning that our grammars can be easily adapted to generate different syntactic or semantic structures.

Further, our dataset is designed for the English language. Some phenomena we test do not appear in all languages; e.g. not all languages can stack adjectives indefinitely. There are many interesting phenomena that are more pronounced in some non-English languages, that we do not test here, for example how a parser would deal with richer morphology.

One possible application of our dataset is not only in the evaluation of published parsers, but also during the development of new parsers. For example, the effect of a change to the parsing architecture, designed to address parsing performance for a specific phenomenon, could be evaluated using GrAPES. However, since many of our examples are drawn from the AMRBank test set, and GrAPES overall has no dev/test split, this can lead to overfitting to the test set. For now, for development we recommend only using datasets from GrAPES that are not drawn from the AMR test set. When reporting results on GrAPES, if some parts of GrAPES were used during development, that fact should be included in the report with high visibility. We hope to publish a development set for GrAPES in the near future.

Despite our efforts to make our metrics focus precisely on the specific tasks, sometimes less relevant errors are caught. For instance, the exact match metric for structural generalization can yield a zero if there are lexical errors. The sanity checks are designed to catch such issues, but will not catch all. For example, an analysis of the AM Parser outputs for Nested Control and Coordination found that the 60% error rate was driven largely by lexical problems linking the word *me* to an i node (in other parser-category pairings we examined, we did find mostly structural errors). Possible fixes could include finding another metric for some structural generalization categories, or perhaps changing the lexical distribution in our grammar-generated corpora.

Given the scale of our dataset, it includes possible annotation errors and surface-form ambiguities, as is the case with most datasets of that scale. Our inspection of the dataset finds that these are minimal, but future work may focus on further cleaning up the dataset or quantifying the level of noise in it. In the case of ambiguity, future work may also create several possible references for each possible reading.

## Ethics Statement

We do not see any particular ethical concerns with this work.

## Acknowledgements

Many thanks go to our supporting annotators: Maria Francis, Christoph Otto, and Anna Spasiano. We would also like to thank Alexander Koller, Matthias Lindemann, Ivan Titov, and Juri Opitz for insightful discussions. Thanks also to Aditya Surikuchi and Sandro Pezelle for feedback on the paper. Last but not least, we would like to thank the reviewers for their helpful comments. This work is funded in part by the Deutsche Forschungsgemeinschaft (DFG, German Research Foundation) – 492792184. This work is also part of the research programme *Learning meaning from structure: neural semantic parsing with minimalist grammars* with project number VI.Veni.194.057, which is funded by the Dutch Research Council (NWO).

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

## A Gender bias observations

Our qualitative evaluation revealed a form of bias in AMRBart for the CP recursion with RC and coreference category. The sentences in this category take the form

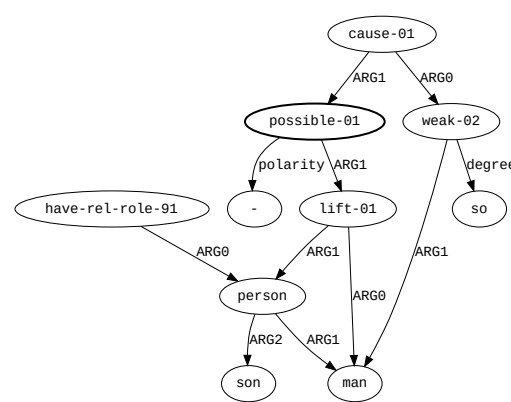

Figure 2: AMR for *The man couldn't lift his son because he was so weak.*

(2)    The kids who [I said] liked the astronaut actually hated her after all.

where the part in square brackets can be a deeper CP recursion. We used other lexical items in place of *astronaut*, that all were common in the dataset, which all turned out to be stereotypically male: *lawyer*, *doctor* and *soldier*. A manual examination of 30 parses by AMRBart showed that, when the underlined pronoun was male (*actually hated him after all*), the parser correctly resolved the coreference 14 out of 15 times. However, when the pronoun was female, as in the example above, AMRBart correctly resolved the coreference only 7 out of 15 times. The difference is statistically significant (two proportion z test, $p = 0.005$). This is indicative of a gender bias in the model.

## B Appendix: Corpus and Evaluation Details

In this Appendix we go through the categories one by one, providing for each a description, examples, and explanations of the evaluations used. A few concepts will come up a few times; their definitions are given below.

### Definitions

We will use the example in Fig. 2 to illustrate the definitions. It is taken from the Winograd corpus (Levesque et al., 2012), and annotated with an AMR by us.

**Node label recall** : there is a node somewhere in the graph with this label.

**The reentrant node** : The node we are interested in that should have two incoming edges. In Fig. 2, this is man, because we are evaluating the ability of the parser to predict that *he* is the man, not the son or someone else outside the sentence.

**Near parent** : For a reentrant node, the "near" parent is near in the sentence, for instance the predicate that selects a pronoun. In Fig. 2, this is weak-02.

**Far parent** : The "far" parent is the parent that "first" introduces the subgraph, for instance the predicate that selects a full RE. "first" here is either highest co-indexed c-commander in the syntax tree (so, often the predicate that selects the binder) or leftmost in the sentence if the former doesn't apply. In Fig. 2, this is lift-01.

### B.1 Pragmatic reentrancies

**Description** Reentrancies that are not forced either by the structure or by the use of 1st or 2nd person pronouns (compare Unambiguous reentrancies). Includes third person pronouns, epithets, repetition of the RE, among others

**Example** *Obama's VP said the president forgot his briefcase.* Here, *the president* is an epithet for Obama, and *his* is a pronoun referring to Obama, but both of these people could in principle be someone other than Obama, so these are pragmatically coreferent.

**Dataset sources**

- Test set
  - extracted automatically; hand-labelled by type of co-reference
- Winograd (Levesque et al., 2012) A Winograd Schema is a pair of sentences that differ only in one or two words and that contain a referential ambiguity that is resolved in opposite directions in the two sentences. The resolution must require pragmatics, as opposed selectional restrictions such as animacy.
  The corpus was created as an alternative to the Turing Test for AI, based on an example from Winograd's dissertation (Winograd, 1971), which comprise the first two sentence of the corpus:

(3) The **city councilmen** refused the demonstrators a permit because **they** feared violence.

(4) The city councilmen refused the **demonstrators** a permit because **they** advocated violence.

**Evaluation**

**Metric** Edge recall (labelled)
- 'near' parent to reentrant node
- 'far' parent to reentrant node

**Senses?** No

**Prerequisites** reentrant node, near parent, far parent

### B.2 Unambiguous reentrancies

Reentrancies with only one possible interpretation, in contrast to set 1 above. There are two kinds, *Syntactic (gap) reentrancies*, in which the structure forces the co-reference, and *Unambiguous coreference* in which the lexical items force the co-reference.

#### B.2.1 Syntactic (gap) reentrancies

**Description** Includes control, nominal control, secondary predication, and coordination

**Example** *She wants and needs to enter the room whistling*: *want* and *need* are control verbs, so they share their ARG0 with *enter*. *wants and needs* is a coordinated VP, sharing *she* as their ARG0. *whistling* is a subject depictive secondary predicate, also with *she* as its ARG0.

**Dataset source** Test set

- extracted automatically; hand-labelled by type of co-reference

**Evaluation**

**Metric** Edge recall (labelled)
- 'near' parent to reentrant node
- 'far' parent to reentrant node

**Senses?** No

**Prerequisites** reentrant node, near parent, far parent

#### B.2.2 Unambiguous coreference

**Description** Includes 1st and 2nd person pronouns, "self".

**Example** *I raised my fists in self-defence*: 1st person pronoun (*I, my*) yield a reentrancy at i; *self* adds another.

**Dataset source** Test set

- extracted automatically; hand-labelled by type of co-reference

**Evaluation**

**Metric** Edge recall (labelled)
- 'near' parent to reentrant node
- 'far' parent to reentrant node

**Senses?** No

**Prerequisites** reentrant node, near parent, far parent

## B.3 Structural Generalization

Increasingly deep recursion of various structures (see below). All of these subcorpora are generated by synchronous grammars. A combination of grammar design and sampling constraints prevents ambiguity. For instance, in grammar design, person and number (dis)agreement is leveraged to keep intervening noun phrases from being interpretable as co-referent with other noun phrases. In sampling, some rules cannot be reused, for instance a noun cannot be repeated so that no reentrancy will arise there.

Care was taken to keep everything except the phenomenon of interest as simple as possible. This includes single-node nouns (somewhat rare in AMR for humans, hence the repeated uses of girl, boy, astronaut, mechanic, soldier, lawyer, etc.) and ensuring that the words are common enough in the training set that our test parsers succeeded on the sanity checks. In this way, we expect that difficulties parsing these sentences reflect difficulty with structural generalisation, not with the building blocks.

### B.3.1 Nested control and coordination

**Description** Control structures within control structures and coordination. Both of these phenomena create unambiguous reentrancies, and here they are nested up to 8 clauses or coordinations deep.

**Example** *The boy wanted to force the doctor to refuse to attend and jumped.* (depth 4)

VP coordination forces the ARG0 of both *wanted* and *jumped* to be *boy*. Meanwhile,

*wanted* is a subject control verb, which causes *boy* to be the ARG0 of *force*, creating another reentrancy at boy. *Force* is itself an object control verb, so its ARG1, *doctor*, is also the ARG0 of *refuse*. *Refuse* is again a subject control verb, creating another reentrancy at doctor.

**Dataset source** Grammar

Care was taken here to avoid structural ambiguity, for example choosing verbs that couldn't be interpreted to take a nominal object.

In the example, notice how the agreement keeps it unambiguous. If the main clause verb were in the first person present, the last word would be *jump*. In this case, it could also be the doctor who jumped.

**Evaluation**

**Metric** Exact match

**Edges Label?** No

**Senses?** No

**Sanity Check** Exact match on unnested control and coordinated non-control verbs, all covering the same vocabulary. e.g. *The boy wanted to jump.*

### B.3.2 Multiple adjectives

**Description** Noun phrases with stacked adjectives up to 5 deep

**Example** *A strange big antique square dark container*

**Dataset source** Grammar

- Ordering restrictions on English adjectives is respected: opinion, size, age, shape, colour, material.
- Nouns and adjectives were carefully chosen to avoid structural ambiguity, in which an adjective could be interpreted as a noun. For instance, we avoid colour names in favour of *pale* and *dark* because, e.g., the substring *strange red* of *a strange red car* could be taken to be a strange colour of red, with nominal modification of *car*.

**Evaluation**

**Metric** Exact match

**Edges Label?** No

**Senses?** No

**Sanity Check** Exact match on single-adjective noun phrases covering the same vocabulary, e.g. *A fantastic plate*.

### B.3.3 Centre embedding

**Description** Recursive relatives modifying the subject, up to depth 4

**Example** *The astronaut who the girl who the boy hugged taught left*

Centre embedding is hard for humans. This example has this structure: *The astronaut who [[the girl who the boy hugged] taught] left*.

**Dataset source** Grammar

**Evaluation**

**Metric** Exact match

**Edges Label?** No

**Senses?** No

**Sanity Check** Sentences without relative clauses covering the same vocabulary, e.g. *The doctor hugged the mechanic*.

### B.3.4 Long lists

**Description** Lists up to length 35, embedded in a simple context. AMR handles coordination with one and node and as many `opi` edges as needed.

**Example** *Please buy a book, gasoline, fish, expensive food, beer, soap, a map, a phone and coal.*

This will require edges op1 through op9.

**Dataset source** Grammar

**Evaluations**

**Metric** Conjunct recall: Evaluates what percentage of conjuncts seen in the gold graph are also conjuncts in the predicted graph.

**Metric** Conjunct precision: Evaluates what percentage of conjuncts seen in the predicted graph are also conjuncts in the gold graph. Since the sample size depends on the number of conjuncts in the predicted graph, i.e. differs by parser, we do not report it here.

**Metric** Unseen `:opi` recall:
for i = 20, 21, . . .

**Senses?** No

**Edge labels?** Only for Unseen `:opi` recall

**Sanity Check** Exact match on sentences with the same context but with only one item, covering the whole vocabulary. e.g. *Please buy fish*; *I saw 98 rats*

### B.3.5 CP recursion

CP recursion is a kind of sentential embedding, in which Complementizer Phrases (usually headed by *that*) embed a sentence as the object of a verb, as in *We said that the astronaut left*.

Sentential embedding can create long-distance dependencies between verbs and their arguments; for example in *The girls who we claimed that you thought slept hated the lawyer*, the subject *the girls* is far away from the main verb *hated*.

In addition to distance in the string, we can talk about distance in the constituency tree: the length of the path between the dependent elements. Here the distance between *the girls* and *hated* is quite short, as the whole relative clause *who we claimed that you thought slept* is just a modifier of *girls*.

GrAPES includes four kinds of CP recursion, corresponding to the full typology of dependency distance in the string and distance in the tree. The first category has no reentrancies or relative clauses, so all dependencies are short in the string and the tree.

**Description** CP recursion with no reentrancies or relative clauses, up to depth 10

**Example** *The lawyer said that you knew that the men mentioned that the boys believed that the women left*

**Dataset source** Grammar

- In sampling from the grammar, no repeated nouns were allowed, to avoid accidental reentrancies.
- All verbs are either CP-selectors like *think* or intransitive.

**Evaluation**

**Metric** Exact match

**Edges Label?** No

**Senses?** No

**Sanity Check** Sentences with one embedded CP, covering all vocabulary

**Notes** All dependencies are short in the string and the syntax tree.

### B.3.6 CP recursion with coreference

**Description** CP recursion with coreference:

- 1st and 2nd person pronouns at a distance of 1 to 10 CPs
- 3rd person pronouns at a distance of 2 to 10 CPs. Third person pronouns have a clarifying noun phrase to make them unambiguous (see example)

**Example** *We heard that **you** mentioned that I said that the kids liked **you**.*

**Example** *I thought that **the doctor** heard that we mentioned that the lawyer mentioned that the girls hated **him**, the doctor*

**Dataset source** Grammar

- In sampling from the grammar, no repeated nouns were allowed except for the target nouns and pronouns, to avoid accidental reentrancies.
- The 3rd person pronoun and clarifying noun phrase always appear at the end, as objects of the last verb.
- First occurrences are not always the first noun phrase in the sentence, and reentrant 1st and 2nd person pronouns are not always last

**Evaluation**

**Metric** Exact match

**Edges Label?** No

**Senses?** No

**Sanity Check** Sentences with one embedded CP, and coreference, covering all vocabulary
- For pronouns: one reentrant pronoun pair
- For 3rd person: One CP, e.g. *The lawyer knew that I hated her, the lawyer.*

**Notes**
- The dependency created by the coreference is long in the string and long in the syntax tree.
- We use 1st and 2nd person pronouns because they are unambiguous, but they can be made easy by merging all i, we, and you nodes in post-processing. Third person pronouns are always ambiguous, but the context here makes any other

choice of referent extremely pragmatically anomalous.
- Note that there are no unambiguous third-person reentrancies at a distance in English – e.g. long-distance reflexives like in Farsi – hence the design of the 3rd person variants.

### B.3.7 CP recursion with relative clause (RC)

**Description** CP recursion appearing within relative clauses modifying subjects. The CPs push the relativised noun farther from its main predicate. We have 1-5 CPs within the relative clause.

**Example** *The **girls** [who we claimed that you thought **slept**]$_{RC}$ **hated** the lawyer*

While there is no cycle in the graph here, there are two incoming edges to the relativised noun *girls*: from the main verb in the sentence *hated* and the deepest embedded verb in the RC, *slept*.

The two embedded CPs *[that you thought [slept]$_{CP}$]$_{CP}$* push *hated* farther from its subject *the girls*.

They also separate the verb *slept* from the dependent relativised noun *girls* and relative pronoun *who*.

**Dataset source** Grammar

**Evaluation**

**Metric** Exact match

**Edges Label?** No

**Senses?** No

**Sanity Check** One CP, e.g. *The boys who I thought sneezed hated the doctor.*

**Notes**
- Both dependencies are long in the string, but the main clause verb is close in the syntax tree.
- Note that in the example, *slept* is in fact a CP. It is missing its subject because of the relativisation, and there is no *who* because of the so-called *that*-trace effect in English.

### B.3.8 CP recursion with RC and coreference

**Description** CP recursion appearing within relative clauses modifying subjects, plus 3rd person pronominal coreference with a strong

pragmatic context. (See example). We have 1-7 CPs within the relative clause.

**Example** *The astronaut [who we said [liked the lawyer]$_{CP}$]$_{RC}$ actually hated **her** after all*

In addition to the long distance dependency in the RC, the pronoun *her* creates a reentrancy at `lawyer`.

**Dataset source** Grammar

- All sentences have the following template:
  *The* N1 *who* [relative clause with CP recursion] *liked/hated the* N2 *actually hated/liked him/her after all*
- *liked* is always paired with *hated* and vice versa

**Evaluation**

**Metric** Exact match

**Edges Label?** No

**Senses?** No

**Sanity Check** One CP, e.g. *The girls who we claimed liked the doctor actually hated him after all.*

**Notes** While all third person pronouns are technically ambiguous, the contrast between *like* and *hate* and the *actually… after all* makes it pragmatically very hard to interpret the pronoun to be anyone other than the object of the earlier *liked/hated*

The reentrancy corresponds to a short dependency in the string but a long one in the syntax tree because the antecedent (the full noun phrase) is deeply embedded in the subject relative clause, but the pronoun is the object of the main verb of the sentence.

## B.4 Rare and unseen words

### B.4.1 Rare node labels

**Description** Node labels in the test set that are seen one to five times in the training set

**Example** `centrifuge`

**Dataset source** Test set

- extracted automatically; hand filtered to remove annotation errors and words inconsistently annotated in the training set

**Evaluation**

**Metric** Node label recall

**Senses?** Yes

**Prerequisites** None

### B.4.2 Unseen node labels

**Description** Node labels in the test set that are not present training set

**Example** `gown`

**Dataset source** Test set

- extracted automatically; hand filtered to remove annotation errors and words inconsistently annotated in the training set

**Evaluation**

**Metric** Node label recall

**Senses?** Yes

**Prerequisites** None

### B.4.3 Rare predicate senses (excl. `-01`)

**Description** PropBank predicate senses that occur fewer than five times in the training set, but whose predicates also occur with other senses in the training set at least once. We exclude the `-01` sense since it is universally very common, making it to guess blindly.

**Example** *Loose tee shirts, Nursing bras, nursing pads.* → `loose-03`

**Dataset sources** Test set

- extracted automatically; hand filtered to remove annotation errors and words inconsistently annotated in the training set

**Evaluation**

**Metric** Node label recall

**Senses?** Yes

**Prerequisites** Node label recall without senses

### B.4.4 Unseen predicate senses (excl. `-01`)

**Description** PropBank predicate senses that do not occur in the training set, but whose predicates occur with another sense in the training set at least once. We exclude the `-01` sense since it is universally very common, making it possible to guess blindly. `unseen_senses_test.tsv`?

**Example** `fill-in-07`

- *The young reporter filled in for the usual news anchors.*
- *The young reporter filled in for the usual news anchors while they were on vacation.*

**Dataset sources** Hand-written

- Sentences appear in pairs, where the second extends the first with more pragmatic cues (see example)
- We found that the test set included too few correct examples, so we created our own corpus.
- We chose the predicates by how easy it was for us to come up with example sentences for them.

**Evaluation**

    **Metric** Node label recall
    **Senses?** Yes
    **Prerequisites** Node label recall without senses

### B.4.5 Rare edge labels (`ARG2+`)

**Description** Argument edge labels (`ARG2-ARG5`) that occur one to five times in the training set with the given predicate.

In the example below, this means `develop-02` occurs one to five times with an `ARG3` edge.

**Example** *we can actually get some commercial development and have the ability to work closer to where we live.* → (`develop-02 :ARG3 we`)

**Dataset sources** Test set

- extracted automatically; hand filtered to remove annotation errors and words inconsistently annotated in the training set

**Evaluation**

    **Metric** Edge recall (labelled)
    **Metric** Edge recall (unlabelled)
       - from the predicate to the argument
    **Senses?** Yes
    **Prerequisites** Both nodes, including Prop-Bank sense

**Notes** We include sense disambiguation on the predicate because the meaning of a role varies with the sense of the predicate.

### B.4.6 Unseen edge labels (`ARG2+`)

**Description** Argument edge labels (`ARG2-ARG5`) that do not occur in the training set with the given predicate.

**Example** *The kids bounced the ball five meters onto the roof.*

```
(bounce-01
    :ARG2 distance-quantity ...
    :ARG4 roof)
```

**Dataset sources** Hand-written

- As with unseen predicate senses, we found that the test set included too few correct examples.

**Evaluation**

    **Metric** Edge recall (labelled)
       - from the predicate to the target
    **Senses?** Yes
    **Prerequisites** Both nodes, including Prop-Bank sense

**Notes**
- Predicates might not occur in the training set at all; the prerequisite test checks for the predicate.
- We include sense disambiguation on the predicate because the meaning of a role varies with the sense of the predicate.

## B.5 Special entities

AMR treats names, dates, and other entities such as URLs, scores, phone numbers, etc specially, with a node labelled with the type of entity and details given in attributes. For example, names have a `name` node with the parts of the name as `opi` attributes. We test on both seen and unseen entities.

### B.5.1 Seen names

**Description** Named entity names that occur in the training set

**Example** *Capitol Hill* → (`n / name :op1 "Capitol" :op2 "Hill"`)

**Dataset sources** Test set

- extracted automatically

**Evaluation**

**Metric** Recall on sequence of `opi` target node labels. If there is no such node, recall is 0.

Names are subgraphs rooted in a `name` node. The components of the name are targets of `opi` edges. The name is correct if there is a `name` node whose `opi` edge target labels match the gold sequence of `opi` edge target labels. In the example above, this is the sequence `Capitol Hill`.

**Prerequisites** None

### B.5.2 Unseen names

**Description** Named entity names that don't occur in the training set

**See seen variant, above, for details**

### B.5.3 Seen dates

**Description** Dates where the same graph fragment occurs in the training data

**Example** *December 22, 2002*

```
(d / date-entity :month 12 :day 22
:year 2002)
```

**Dataset sources** Test set

- extracted automatically

**Evaluation**

**Metric** Recall on sequence of attributes of `date-entity`. If there is no such node, recall is 0.

In this example above, this is `day 22 month 12 year 2002`

**Prerequisites** None

### B.5.4 Unseen dates

**Description** Dates where the graph fragment does not occur in the training data (some parts of the fragment, such as the month or the year, may occur in the training data in a different date).

**See seen variant, above, for details**

### B.5.5 Other seen entities

**Description** Other special entity types (numerical values, phone numbers, string literals, urls, . . . ), where that value has occurred in the training data.

**Example** *...please call [him] on his cell: 470-5715...*

```
:value "470-5715"
```

**Dataset sources** Test set

- extracted automatically

**Evaluation**

**Metric** Recall on the value (here: `"470-5715"`)

**Senses?** N/A

**Prerequisites** None

**Notes** Full list of possible entities: `"data-entity"`, `"percentage-entity"`, `"phone-number-entity"`, `"email-address-entity"`, `"url-entity"`, `"byline-91"`, `"correlate-91"`, `"course-91"`, `"have-degree-of-resemblance-91"`, `"hyperlink-91"`, `"instead-of-91"`, `"publication-91"`, `"request-confirmation-91"`, `"score-entity"`, `"score-on-scale-91"`, `"statistical-test-91"`, `"street-address-91"`, `"string-entity"`, `"value-interval"`, `"variable"`

### B.5.6 Other unseen entities

**Description** Other special entity types (numerical values, phone numbers, string literals, urls, . . . ), where that value has not occurred in the training data.

**See seen variant, above, for details**

## B.6 Entity classification and linking

Named entity types and their wiki links.

### B.6.1 Types of seen named entities

**Description** Entities that occur in the training set. We test recall on the type (e.g. `person`, `company`, `canal`)

**Example** *County and city officials in Los Angeles tried to determine...* → `city`

**Dataset sources** Test set

- extracted automatically

- The type of a named entity is the `:name` parent of any `name` node. In the example, we get `city` from `(c / city :wiki "Los_Angeles" :name (n / name :op1 "Los" :op2 "Angeles"))`

**Evaluation**

**Metric** Node label recall

**Prerequisites** There is a `name` node in the graph with attributes, in order, the same as in the gold graph.

In the example, we look for one with exactly two attributes, `Los` and `Angeles`, in that order.

**Notes** In AMR, the type is taken from the text if present; otherwise one is taken from the list of standard NE types. See `https://github.com/amrisi/amr-guidelines/blob/master/amr.md#named-entities`

### B.6.2 Types of unseen named entities

**Description** Entities that do not occur in the training set. We test recall on the type (e.g. `person`, `company`, `canal`)

**See seen variant, above, for details**

### B.6.3 Seen and/or easy wiki links

**Description** Wiki links for entities that occur, with their wiki links, in the training set, or whose wiki links are just the name joined with underscores, e.g. *Barack Obama*'s wiki link is just `Barack_Obama`.

**Example** *North Korean officials refused to proceed...* → `"North_Korea"`

**Dataset sources** Test set

- extracted automatically
- The Wiki link of a named entity is any `:wiki` attribute of any node.
  In the example, we get `"North_Korea"` from `(c / country :wiki "North_Korea" :name (n / name :op1 "North" :op2 "Korea")`, and it qualifies as easy because `"North_Korea"` is just the `op`is in order, `"North"` and `"Korea"`, joined with a `"_"`.

**Evaluation**

**Metric** Node label recall
If the graph contains a `:wiki` edge with the correct target, it counts as correct.

**Prerequisites** None

### B.6.4 Hard unseen wiki links

**Description** Wiki links for entities that do not occur in the training set, **and** whose Wiki links are not just the name joined with underscores, e.g. the way *Barack Obama*'s wiki link is just `Barack_Obama`.

**Example** *Police sources also intimated that more crackdowns by the Hong Kong Police on other Triad gangs will follow.* → `"Triad_(organized_crime)"`

**Dataset sources** Test set

- extracted automatically; filtered to exclude annotation errors
- The Wiki link of a named entity is any `:wiki` attribute of any node.
  In the example, we get `"Triad_(organized_crime)"` from `(c3 / criminal-organization :wiki "Triad_(organized_crime)" :name (n2 / name :op1 "Triad"))`, and it qualifies as hard because `"Triad_(organized_crime)"` is not just the name, `"Triad"`.

**Evaluation**

**Metric** Node label recall
If the graph contains a `:wiki` edge with the correct target, it counts as correct.

**Prerequisites** None

## B.7 Lexical disambiguations

Words in the sentence do not always have straightforward annotations in the graph. Here we test PropBank predicate senses (e.g. the `-01` of `use-01`) and words with ambiguous meanings such as *like*, which can mean both *similar to* and *enjoy*, as in *Time flies **like** an arrow vs Fruit flies **like** a banana.*

### B.7.1 Frequent predicate senses

**Description** PropBank senses that occur at least times in the training data

**Example** *If he really loves his son, he'll see him regardless, but this is just a typical control method that people like him use.* `use-01`

`use-01`

**Dataset sources** Test set

- extracted automatically; filtered automatically to exclude `include-01` and `include-91` as they are inconsistently annotated in the training set.

**Evaluation**

**Metric** Node label recall, including senses (e.g. `use-01`)

**Prerequisites** Node label recall, excluding senses (e.g. `use`)

### B.7.2 Other word ambiguities

**Description** Words with multiple meanings.

**Example** *Time flies **like** an arrow. Fruit flies **like** a banana*

**Dataset sources** Two data sets are evaluated on, an existing non-AMR corpus and a handwritten corpus, which also contains 12 test set sentences.

1. Putting Words into BERT's Mouth (Karidi et al., 2021)

   - Sets of sentences for ambiguous words *had, in, on, run, started, with, about, for*
   - Not all senses of the words are annotated differently in AMR, so from the perspective of AMR, the dataset is not entirely balanced, but from a linguistic perspective, it is.
   - e.g. for ambiguous word *in*,
     *The event is in Canada.* → `be-located-at-91`
     *The event is in June.* → `be-temporally-at-91`
   - Hand-annotated with full AMRs (provided as a supplementary data set in our repository)

2. Handwritten (plus 12 from the test set)

   - Chosen to complement Karidi et al. (2021)
   - Contains the following words and meanings:

| word | Subgraph |
|------|----------|
| *as* | `:time` |
|      | `cause-01` |
| *like* | `resemble-01` |
|      | `like-01` |
| *really* | `:degree really` |
|      | `real-04` |
| *since* | `since` |
|      | `cause-01` |
| *over* | `:time over` |
|      | `:location over` |

- About 6 sentences per word sense
- All of the sentences are either unambiguous or are pragmatically strongly favoured in the intended way. However, we also made a point of making many of the latter technically ambiguous, even if it requires some serious mental gymnastics. For instance, perhaps fruit does fly like a banana flies.
- Every word sense has one or two entries hand-selected from the test set. These are included in the corpus without their AMRs since the dataset has restricted access. The test set IDs are listed below.

**Evaluation**

**Metric** Node or edge recall
  - Some are non-core roles, and we accept both the edge and the reification

**Edges Label?** Yes

**Senses?** Yes

**Prerequisites** None

**Notes** Test set entries:

```
PROXY_NYT_ENG_20081128_0005.6
DF-199-194215-653_0484.4
DF-199-194215-653_0484.9
PROXY_LTW_ENG_20070930_0021.32
DF-200-192400-625_7806.1
DF-200-192400-625_6304.24
DF-200-192400-625_6304.9
PROXY_NYT_ENG_20050716_0171.12
PROXY_XIN_ENG_20040429_0189.23
NW_AFP_ENG_0013_2003_0427.4
PROXY_XIN_ENG_20041010_0024.5
NW_XIN_ENG_0209_2008_0513.21
```

### B.8 Edge attachments

A misplaced edge can dramatically change the meaning of a sentence. Here we examine four kinds of edge attachment: PP attachment, in which a prepositional phrase can attach to a noun or a verb, but one is strongly favoured over the other; unbounded dependencies, such as long-distance wh-extraction, relatives clauses, and right node raising; and passives and unaccusatives, in which the syntactic subject of the sentence is the ARG1 of the predicate.

#### B.8.1 PP attachment

**Description** Specifically designed sentences to test PP attachment ambiguities. If one ignores the lexical content, just structurally speaking, the PP can attach to the VP or the NP, but with the lexical content, there is either only one option licensed by selectional restrictions, or a strong semantic/pragmatic preference.

**Example** *Sophie knew the journalist with the telescope*

Check for an edge `telescope :poss journalist` or a subgraph `(have-03 :ARG0 journalist :ARG1 telescope` or `(own-01 :ARG0 journalist :ARG1 telescope`

**Dataset source** Grammar

- Grammars are designed so that incorrect attachments are ruled out by selectional restrictions; for instance, *knew* is incompatible with a seeing instrument like *telescope*. In other sentences, both readings are possible, but one much more likely. For example, in *The man sees the moon with the telescope*, in principle, the moon could have a telescope, but it is much more likely to be the seeing instrument.
- see Table 8 for templates
- vocabulary was chosen to be present in the training set and to be recognised by AMRBart and AM Parser.

**Evaluation**

**Metric** Edge recall (labelled)
- Edge between the root of the modified subgraph and the root of the modifier, e.g. between `telescope` and `journalist`.

- Most of these PPs have reifications, so both are accepted; e.g. `own-01`, `have-03` for `:poss`

**Senses?** Only for the reification nodes, if they exist.

**Prerequisites** Since we are looking for an edge or a reification subgraph connecting two nodes, the prerequisite is the presence of both nodes, modulo senses.

#### B.8.2 Unbounded dependencies Rimell et al. (2009)

**Description** The Unbounded Dependencies corpus consists of sentences drawn primarily from the Penn Tree Bank, annotated with two dependent words and a type of dependency. Some dependencies entail reentrancies, and some only two incoming edges to the same node. They are considered unbounded because there is no limit on how far apart, in the string and in the constituency tree, the two elements can be. Types are:

1. Object extraction from a relative clause
2. Object extraction from a reduced relative clause
3. Subject extraction from a relative clause
4. Free relative
5. Object wh-question
6. Right node raising
7. Subject extraction from an embedded clause

**Example** *That finished the **job** that Captain Chandler and Lieutenant Carroll had **begun**.*

This is an object relative, and we look for an edge between the embedded predicate *begun* and the relativised noun *job*: `(begin :ARG1 job)`

**Dataset source** Unbounded dependencies Rimell et al. (2009)
- This is not an AMR corpus
- Hand-annotated with relevant nodes and the edges between them.

**Evaluation**

**Metric** Edge recall (labelled)
- Edge between roots of subgraphs with the meanings of the two dependent words

| Verb | Object | P | NP-modifiers | VP-modifiers |
|---|---|---|---|---|
| gave up, abandoned | her ambitions, aspirations, career, dreams | **in** | mathematics, theatre, crime | anger, a fit of despair, a moment of clarity, the 60s, july, 2012, spring |
| Example: | *My sister gave up her ambitions in mathematics*
*My sister gave up her ambitions in a fit of despair* | | | |
| bought, acquired, purchased, picked up | onions, mushrooms, tomatoes, carrots | **for** | the pasta sauce, the salad, the soup | $5, $10, a few dollars, almost nothing, an unreasonable amount of money |
| Example: | *Kim bought mushrooms for the soup*
*Kim bought mushrooms for $5* | | | |
| has kept | this postcard, letter, necklace, souvenir | **from** | Minsk, Munich, that adventure, Haiti | |
| has kept | this information, knowledge, news, wisdom | **from** | | the children, Mark, the police, Jenny |
| Example: | *For thirty years, she has kept this letter from Minsk*
*For thirty years, she has kept this news from Mark* | | | |
| read, skim, devour | this book, essay, novel | **by** | Barack Obama, J. K. Rowling, Charles Dickens, this young author | tomorrow, tonight, Monday, Tuesday, candlelight, firelight, lamplight |
| Example: | *I will read this book by Charles Dickens*
*I will read this book by tonight* | | | |
| saw, looked at, peeked at, observed | the girl, stranger, soldier, journalist | **with** | the hat, red T-shirt, weird hair, large eyebrows | |
| saw, looked at, peeked at, observed | the northern lights, moon, rainfall, army | **with** | | the telescope, binoculars, spyglass |
| understood, knew, hated, sang to, addressed | the girl, stranger, soldier, journalist | **with** | the telescope, binoculars, spyglass | |
| Example: | *The baker saw the stranger with the weird hair*
*The baker hated the stranger with the telescope*
*The baker saw the northern lights with a telescope* | | | |

Table 8: PP attachment templates

- The dependent words are the words annotated in the corpus to have a dependency

**Senses?** No

**Prerequisites** Existence of both nodes that should be connected with an edge. In the example, begin and job

### B.8.3 Passives

**Description** Passive sentences without an ARG0 (so without a *by*-phrase)

**Example** *Kimball stated the Iranian government's design is to deflect criticism and pressure and to claim that **progress is being made***

```
(make-01 :ARG1 progress-01)
```

**Dataset sources** Test set

- extracted automatically based on predicates with no ARG0
- Hand-filtered and categorised to separate unaccusatives (see below), and to remove errors and other constructions such as adjectives and expletives.
- Sentences with *by*-phrases aren't included because automatic extraction is much easier if you're just looking for predicates with no ARG0.

**Evaluation**

**Metric** Edge recall (labelled)

    on ARG1 edge

**Senses?** Yes

**Prerequisites** Existence of both nodes that should be connected with the edge.

### B.8.4 Passives

**Description** Unaccusatives: predicates that don't have an ARG0 in their PropBank frame at all

**Example** *Even though **your anxiety levels would increase** for a duration, they will gradualy decrease so hopefully, overtime your OCD will disappear (:*

`(increase-01 :ARG1 (l / level)`

**Dataset sources** Test set

- extracted automatically based on predicates with no ARG0
- Hand-filtered and categorised to separate passives (see above), and to remove errors and other constructions such as adjectives and expletives.
- includes double unaccusatives, which have more than one ARGi but no ARG0. We only evaluate the ARG1 on these.

**Evaluation**

**Metric** Edge recall (labelled)

    on ARG1 edge

**Senses?** Yes

**Prerequisites** Existence of both nodes that should be connected with the edge.

### B.9 Nontrivial Word to Node Relations

### B.9.1 Ellipsis

**Description** Sometimes a word in the sentence is depicted twice in the graph, as for example with ellipsis. For instance, in the example below, there are two write-01 nodes.

**Example** *Mary wrote a paper and Susan did too.*

**Dataset sources** Test set

- extracted automatically using an alignment tool, LEAMR (Blodgett and Schneider, 2021). Entries with a a word aligned to two nodes with the same label were extracted, and the sample filtered by hand to remove alignment and annotation errors.

**Evaluation**

**Metric** Node recall

- 2+ occurrences of node label

**Senses?** Yes

**Prerequisites** One occurrence of the node label

**Notes** AMR is abstract, so naturally the alignments are not a part of the corpus.

### B.9.2 Multinode word meanings

**Description** Some words' meanings comprise more than one node in the graph

**Example** *Teacher* (person :ARG0-of teach-01)

**Dataset sources** Test set

- Extracted automatically using an alignment tool, LEAMR (Blodgett and Schneider, 2021)
- Entries with a word aligned to two nodes with different labels were extracted
- Filtered by hand to remove alignment and annotation errors.

**Evaluation**

**Metric** Subgraph recall

**Edges Label?** Yes

**Senses?** Yes

**Prerequisites** None

**Notes** AMR is abstract, so naturally the alignments are not a part of the corpus.

### B.9.3 Imperatives

**Description** Imperatives have an imperative mode, and most have a (usually latent) you or we (for exhortatives, as in the example). Some also have a different overt subject, as in the second example

**Example**
- *Let's go!*

    `(go-02 :mode imperative :ARG0 we)`
- *Go, Roughriders!*

    `(go-31) :mode imperative :ARG0 (team...))`

**Dataset sources** Test set

- extracted automatically: all entries with an :mode imperative attribute.

- The few entries without a *you* or *we* argument to the predicate were hand-filtered to ensure the correct argument is checked (like in *Go, Roughriders!*)

**Evaluation**

**Metric** Edge recall
- predicate `:mode imperative`
- predicate to you or we

**Edges Label?** Yes

**Senses?** Yes

**Prerequisites** predicate (in the examples, `go-02` and `go-31`)