# OpenReview forum: "AMR Parsing is Far from Solved: GrAPES, the Granular AMR Parsing Evaluation Suite"
_EMNLP/2023/Conference — EMNLP 2023 Main_

### Official Review · Reviewer_KxKy · 2023-08-04

**Soundness:** 3

**Excitement:**

4: Strong: This paper deepens the understanding of some phenomenon or lowers the barriers to an existing research direction.

**Paper Topic And Main Contributions:**

This paper introduces the Granual AMR Parsing Evaluation Suite (GrAPES):
* An evaluation suite (corpus) is contributed for a wide range of phenomena, composed of AMR graphs from AMRBank 3.0 and of newly annotated sentences from a number of other corpora to account for phenomena that are not represented in AMRBank
* Evaluation metrics and tools for analysis
* An evaluation of recent parsers like AMRBart


**Questions For The Authors:**

Question A: Can you comment on the availability of Grapes for other researchers. How, when and under which kind of licensing will it be published?
Question B: Will the analysis software be published as well?

**Reasons To Accept:**

* As the study from Opitz and Frank shows, the usual Smatch evaluation does not necessarily reflect human evaluation. Having having this resource will be very valuable for work on AMR parsing (and beyond), to understand the performance of parsers in detail.
* The choice of phenomena is generally well founded and explained with a lot of detail in the appendix
* The creation of the corpus represents a sizeable effort, and the analysis tools represent an additional value
* Effort of standardizing the evaluation procedures for AMR in future work


**Reasons To Reject:**

* The value of this work for further research depends on the availability of the test suite, and unless I missed something, there do not seem to be any remarks about the availability in the paper.


**Reproducibility:**

5: Could easily reproduce the results.

**Reviewer Confidence:**

3: Pretty sure, but there's a chance I missed something. Although I have a good feel for this area in general, I did not carefully check the paper's details, e.g., the math, experimental design, or novelty.

**Typos Grammar Style And Presentation Improvements:**

* Every phenomenon is evaluated using an appropriate metric, which is ok. However, one could argue that it is problematic that in tables with the results, scores from different metrics computed over different numbers of datapoints are shown/mixed together, or even averaged over (in Table 6), even though a relative comparison is not possible. I think this point should be mentioned.

---

> ### Author Rebuttal · Authors · 2023-08-28
>
> We thank the reviewer for their kind comments.
>
> We will of course make the evaluation suite available open source (on GitHub with a GPLv3 license). We simply forgot to mention this in the original draft, sorry! We will publish the dataset, the evaluation code, the software for visualization and manual analysis of parser errors, and the grammars as well as the code we used to generate sentences from the grammars.
>
> There is one caveat to this: GrAPES relies in part on the AMRBank testset, which is distributed by the LDC (and we thus cannot distribute it ourselves). Thus, to make GrAPES work, the user will need to obtain the AMRBank from the LDC. But since the AMRBank is the standard training and evaluation dataset for AMR parsing anyway, we don’t expect this to be much of a problem for those interested.
>
> Thank you also for the point about showing different metrics in one table or even averaging them. Since the metrics are all on the same scale (0-100) and we specify the metric in each row of Tables 2-5, we felt that this was justifiable. We will be more explicit about it in the final paper, thank you for the feedback.

---

### Official Review · Reviewer_Pruy · 2023-08-04

**Soundness:** 3

**Excitement:**

2: Mediocre: This paper makes marginal contributions (vs non-contemporaneous work), so I would rather not see it in the conference.

**Paper Topic And Main Contributions:**

This paper presents an evaluation suite that tests AMR parsers on a range of phenomena with 36 categories grouped into 9 sets and fine-grained evaluation metrics by category. The suite is designed to answer the questions of how well parsers can handle pragmatic coreference, ellipsis, PP attachment, rare words, unseen named entities and structural generation. The authors apply the proposed evaluation suite to three AMR parsers and discuss evaluation results and provide statistical analysis.

It is not clear to what degree the proposed fine-grained evaluation metrics overlap with previously proposed metrics. For instance, (Damonte et al., 2017; https://arxiv.org/pdf/1608.06111.pdf) evaluates 9 sub-categories including re-entrancies (mostly capturing coreference), named entity and Wikification. In addition (Szubert et al., 2020; https://aclanthology.org/2020.findings-emnlp.199/) discuss re-entrancy phenomena in depth with tools to identify them (https://github.com/mdtux89/amr-reentrancies), which include ellipsis. Rare word and out-of-vocabulary analyses are part of standard error analysis, as shown in (Sheth et al., 2021; https://aclanthology.org/2021.eacl-main.30.pdf).  Structural generation for CP recursion may be a subcase of the challenge posed by parsing long input texts.

It will be more informative if the authors compare and contrast the current proposal against previously published tools and metrics and demonstrate to what extent the current contributions are unique and cannot be captured by any of the previous work.

**Questions For The Authors:**

Lines 398-404: There are major updates in AMR3.0 from AMR2.0. Specifically, frames for superlative and comparatives are subsumed under have-degree-91 and have-qaunt-91 in AMR3.0 that do not exist in AMR2.0 treebank annotations. have-degree-of-resemblance-91 covers constructions such as "(He) is more like (his mother) than (his father)." and correlate-91 covers constructions such as "The more I practised, the fewer mistakes I made.

An easy of way of identifying the difference between AMR3.0 and AMR2.0 test set would be extract the AMR graphs corresponding to identical input texts between AMR2.0 and AMR3.0 and measure Smatch score between the two.

Line 467 & 482-484: To what extent structural generalization can be subsumed under the issue related to long input texts? Which aspect of structural generalization can be decoupled from the input length issue?

**Reasons To Accept:**

The proposed test suite and evaluation metrics can be useful tool for AMR parsing error analysis. But I see quite a few overlap between the current proposal and previous work. And there can be simpler diagnostics than elaborate linguistic analysis, e.g. input length vs. structural generation for CP recursion.

**Reasons To Reject:**

It is not clear to what degree the proposed fine-grained evaluation metrics overlap with previously proposed metrics. For instance, (Damonte et al., 2017; https://arxiv.org/pdf/1608.06111.pdf) evaluates 9 sub-categories including re-entrancies (mostly capturing coreference), named entity and Wikification. In addition (Szubert et al., 2020; https://aclanthology.org/2020.findings-emnlp.199/) discuss re-entrancy phenomena in depth with tools to identify them (https://github.com/mdtux89/amr-reentrancies), which include ellipsis. Rare word and out-of-vocabulary analyses are part of standard error analysis, as shown in (Sheth et al., 2021; https://aclanthology.org/2021.eacl-main.30.pdf).  Structural generation for CP recursion may be a subcase of the challenge posed by parsing long input texts.

It will be more informative if the authors compare and contrast the current proposal against previously published tools and metrics and demonstrate to what extent the current contributions are unique and cannot be captured by any of the previous work.

**Reproducibility:**

2: Would be hard pressed to reproduce the results. The contribution depends on data that are simply not available outside the author's institution or consortium; not enough details are provided.

**Reviewer Confidence:**

5: Positive that my evaluation is correct. I read the paper very carefully and I am very familiar with related work.

---

> ### Author Rebuttal · Authors · 2023-08-28
>
> We thank the reviewer for their detailed comments. If accepted, we will make the distinction to previous work more clear. We believe that our work contains major improvements over the previous state of the art of AMR evaluation, and want to take the opportunity to make this argument here.
>
> The closest predecessor to our work is Damonte et al. (2017). Indeed, Damonte et al. (and its limitations) were a major inspiration for our work. We improve on it in the following ways:
> * As we write in the paper, our work is much more comprehensive (36 vs 9 categories). Getting a more detailed view matters, because often, challenges posed by different phenomena need different solutions, and should thus be evaluated separately.
>    * Example: We split the evaluation of predicted Wikipedia links into one category that looks at links that can be predicted from the text alone, and one that looks at links that need information from actual Wikipedia to be predicted correctly – two tasks that may require very different solutions. We show that the latter is much more difficult, something that can not be seen from Damonte et al.’s evaluation (or any other).
> * In addition to using the existing test set to evaluate parsers on different phenomena, we provide a large amount of new test items, drawn from existing non-AMR corpora, hand-written grammars, and additional sentences of our own.
> * Our metrics specifically evaluate the "tail" of the distribution, difficult phenomena where things still go very wrong with current parsers. Hence, our evaluation suite doubles as a parsing challenge, which previous work does not nearly to this extent.
>   * Example: Results of Damonte et al.’s metric for word sense disambiguation (WSD) imply that current parsers are strong at this task. We confirm this for frequent word senses, but demonstrate the parsers’ weak performance on rare (<50% recall) and unseen (~0% recall) word senses.
> * One challenge in evaluating specific phenomena is to isolate the measurement of that phenomenon from overall performance. Damonte et al.’s metrics only partially succeed in this; we greatly improve on this aspect through our more precise metrics and our method of measuring “prerequisites”.
>   * Example: Take our results on the Winograd subset of GrAPES. We measure coreference resolution here, specifically the existence of a reentrant edge E from a node A to a node B. For us to be able to measure the existence of E, the nodes A and B must exist in the first place – this is what we measure in our “prerequisites” metric. On this dataset, the AMParser obtains 2% recall, with 78% correct prerequisites. This shows that the parser cannot do this task well, and the high prerequisites score shows us that indeed the edges are the problem, not the nodes. By contrast, Damonte et al.’s reentrancy metric gives the AM parser a score of 55 (out of 100) on this dataset, erroneously implying a reasonably strong performance on this difficult task. This is because Damonte et al.’s metric measures all edges at a node that has a reentrancy, rather than measuring the one specific reentrant edge (the latter is only possible with our manual annotations).
> * Damonte et al.’s metrics rely on the AMR test set, which has issues relating to ambiguities and annotation errors when it comes to evaluating high accuracy parsers (shown by Opitz and Frank 2022, and confirmed by our experience). We take great care to circumvent these issues in our work.
>
> The reentrancy metrics in Szubert et al. (2020) are similarly fine-grained as ours, but:
> * They do not measure accuracy directly, but measure how much the overall evaluation score would improve under certain oracle-guided changes to the predicted graph.
> * They divide their metric by oracle-patterns, not by linguistic categories like us.
>
> We think that this makes our metrics more intuitive and useful, but we acknowledge that for reentrancies specifically, our work could be seen more as complementary to Szubert et al. But note that this makes up only 3 out of our 36 categories.
>
> Thank you for bringing the work of Sheth et al. (2021) to our attention. As far as we can see, they only evaluate rare word / OOV properties of intermediate texts, not of the predicted AMRs themselves. To the best of our knowledge, we are the first to evaluate the ability of parsers to predict rare/OOV node and edge labels – and we do it comprehensively, in 11 categories.
>
> Concerning the difference between structural generalization and sentence length: we evaluate several different categories of structural generalization that have similar sentence lengths but vastly different parser results (for example, AMRBART achieves 15% exact match on CP recursion within a relative clause, but 63% exact match on simple CP recursion). This shows that sentence length is not the only factor here. If the paper is accepted, we will clarify the relation between structural generalization and sentence length in the paper.
>
> Overall, we believe that even compared to all existing metrics taken together, our evaluation suite is much more comprehensive, intuitive, precise and challenging, and we hope we convince you as well.

---

### Official Review · Reviewer_qBgp · 2023-08-07

**Soundness:** 4

**Excitement:**

4: Strong: This paper deepens the understanding of some phenomenon or lowers the barriers to an existing research direction.

**Paper Topic And Main Contributions:**

This paper presents an evaluation suite for AMR parsing. The authors first describe the categories on which AMR parsers are evaluated on. They create 36 categories grouped into 9 sets motivated from practical, and linguistic perspectives. Then the evaluation metrics designed are explained. The authors then take three state-of-the-art AMT parsers and evaluate them different categories. Results show that though these parsers achieve great average scores, they perform poorly on some of the categories showcasing the uses to improve up on.

**Reasons To Accept:**

Most of the parsers are evaluated on standard metric and optimized for that metric. This is paper shows the need to look at different aspects of the parser output rather than single metric. Experiments, results and analysis is very thorough. Results of each category are presented with detailed analysis which gives insights in the drawbacks of the parser and areas for improvement. This is an interesting work and ACL community will benefit from analysis like this.

**Reasons To Reject:**

It would be good to mention how this work can be adapted for other grammatical formalisms like CFGs, CCGs etc.

**Reproducibility:**

3: Could reproduce the results with some difficulty. The settings of parameters are underspecified or subjectively determined; the training/evaluation data are not widely available.

**Reviewer Confidence:**

3: Pretty sure, but there's a chance I missed something. Although I have a good feel for this area in general, I did not carefully check the paper's details, e.g., the math, experimental design, or novelty.

---

> ### Author Rebuttal · Authors · 2023-08-28
>
> We thank the reviewer for their thoughtful comments. Thank you especially for your question about how our evaluation suite could be generalized to other syntactic or semantic formalisms. As the grammars we wrote are written in Alto (see citation Gontrum et al. 2017 in the paper), a program specifically designed for supporting multiple formalisms, they can be adapted and re-used for formalisms other than AMR. Additionally, the linguistic categories we use as inspiration for our evaluation may serve as templates for evaluating parsers and representational capabilities of other formalisms along the same lines.
>
> In terms of transferring our actual implementation and annotation work to other formalisms, much of our work is formalism specific: the scripts we used to extract relevant sentence-graph pairs from the AMRBank testset are mostly AMR specific; and the manual filtering and AMR annotation work we did would need to be replicated for any formalism in question.
>
> That said, we hope our work will serve as inspiration for the creation of similar evaluation suites for other formalisms as we believe the general idea will transfer well. We will use some of the extra space in the final version of the paper to detail the above response if accepted.

---

### Meta-Review · Area_Chair_xyzX · 2023-09-17

**Recommendation:** 5

**Metareview:**

This paper introduces a comprehensive new evaluation suite for Abstract Meaning Representation (AMR) parsing, and shows that the AMR parsing remains far from solved.

Pros:
- Thorough evaluation suite covering a range of phenomena, with a practical taxonomy informed by real-world applications.
- Very clearly written, well-structured paper.
- Thorough experiments written up with clear analyses, rich detail.
- Introduces not just a data set, but also standard tooling to compare AMR parsers.
- Data and tools will be publicly available. Clear reproducibility.

Cons:
- Review copy isn't super clear in some parts when it comes to some comparisons with previous work, but if accepted, authors will address this, as per the review thread; as can be seen there, they already have a good understanding of what needs to be added.

This is an exciting contribution to the field; many thanks to the authors.

---

### Decision · Program_Chairs · 2023-10-07

**Decision:**

Accept-Main

**Comment:**

This paper introduces a comprehensive new evaluation suite for Abstract Meaning Representation (AMR) parsing, and shows that the AMR parsing remains far from solved.

Pros:
- Thorough evaluation suite covering a range of phenomena, with a practical taxonomy informed by real-world applications.
- Very clearly written, well-structured paper.
- Thorough experiments written up with clear analyses, rich detail.
- Introduces not just a data set, but also standard tooling to compare AMR parsers.
- Data and tools will be publicly available. Clear reproducibility.

Cons:
- Review copy isn't super clear in some parts when it comes to some comparisons with previous work, but if accepted, authors will address this, as per the review thread; as can be seen there, they already have a good understanding of what needs to be added.

This is an exciting contribution to the field; many thanks to the authors.